# Acute Exposure to Two Biocides Causes Morphological and Molecular Changes in the Gill Ciliary Epithelium of the Invasive Golden Mussel *Limnoperna fortunei* (Dunker, 1857)

**DOI:** 10.3390/ani13203258

**Published:** 2023-10-19

**Authors:** Amanda Maria Siqueira Moreira, Erico Tadeu Fraga Freitas, Mariana de Paula Reis, Júlia Meireles Nogueira, Newton Pimentel de Ulhôa Barbosa, André Luiz Martins Reis, Afonso Pelli, Paulo Ricardo da Silva Camargo, Antonio Valadão Cardoso, Rayan Silva de Paula, Erika Cristina Jorge

**Affiliations:** 1Laboratório de Biologia Oral e do Desenvolvimento, Instituto de Ciências Biológicas, Universidade Federal de Minas Gerais (UFMG), Belo Horizonte 31270-901, MG, Brazil; amandamsmoreira@outlook.com (A.M.S.M.); jumeirelesn@gmail.com (J.M.N.); rayansdpaula@gmail.com (R.S.d.P.); 2Centro de Bioengenharia de Espécies Invasoras de Hidrelétricas (CBEIH), Cidade Nova, Belo Horizonte 31035-536, MG, Brazil; efragafr@mtu.edu (E.T.F.F.); mapreis@gmail.com (M.d.P.R.); newtonulhoa@gmail.com (N.P.d.U.B.); afonso.pelli@uftm.edu.br (A.P.); pricardocamargo826@gmail.com (P.R.d.S.C.); antonio.cardoso@uemg.br (A.V.C.); 3Electron Optics Facility, Materials Science and Engineering, Michigan Technological University, Houghton, MI 49931, USA; 4Center for Population Genomics, Garvan Institute of Medical Research, Sydney, NSW 2010, Australia; andreluiz2207@gmail.com; 5Biotério Nico Nieser, Universidade Federal do Triângulo Mineiro, Uberaba 38025-100, MG, Brazil; 6Escola de Design, Universidade do Estado de Minas Gerais (UEMG), Belo Horizonte 30140-091, MG, Brazil

**Keywords:** biocides, gill epithelium, golden mussel, histology, invasive mollusks, *Limnoperna fortunei*, molluscicide, stress biomarkers

## Abstract

**Simple Summary:**

*Limnoperna fortunei,* known as the golden mussel, is an invasive bivalve in South America responsible for economic damages in hydroelectric plants due to its ability to attach and grow on the walls of water pipes and tubes. Chemical compounds are often used to control golden mussel infestations in water systems. Here, we investigated the effects of two biocides—MXD-100 ™ and sodium dichloroisocyanurate (NaDCC)—through the assessment of morphological and gene-expression alterations in the gills. Both biocides were able to modulate the expression of defensive genes and morphological changes. Yet, it seems that NaDCC needs continuous exposure to control golden mussel infestation, whereas MXD-100™ can inflict severe damage to the mussels in as little as 24 h. Our results can be used to enhance control strategies for managing mussel growth in water systems, simultaneously reducing the environmental impact associated with costly and harmful chemical releases.

**Abstract:**

*Limnoperna fortunei*, the golden mussel, is a bivalve mollusk considered an invader in South America. This species is responsible for ecological and economic damages due to its voluminous fouling capability. Chemical biocides such as MXD-100™ and sodium dichloroisocyanurate (NaDCC) are often used to control *L. fortunei* infestations in hydraulic systems. Thus, we proposed to investigate the effects of different periods (24, 48 and 72 h) of exposure to MXD-100™ (0.56 mg L^−1^) and NaDCC (1.5 mg L^−1^) on the gills of *L. fortunei* through morphological and molecular analyses. NaDCC promoted progressive morphological changes during the analyzed periods and only an upregulation of *SOD* and *HSP70* expression during the first 24 h of exposure. MXD-100™ led to severe morphological changes from the first period of exposure, in addition to an upregulation of *SOD, CAT, HSP70* and *CYP* expression during the first 24 h. In contrast, MXD-100™ led to a downregulation of *CAT* transcription between 24 and 48 h. In static conditions, NaDCC causes lethal damage after 72 h of exposure, and that exposure needs to be continuous to achieve the control of the species. Meanwhile, the MXD-100™ treatment presented several effects during the first 24 h, showing acute toxicity in a shorter period of time.

## 1. Introduction

*Limnoperna fortunei* (Dunker, 1857), known as the golden mussel, is a small freshwater bivalve (subclass Pteriomorphia, order Mytiloida, family Mytilidae) native to the Pearl River basin in China. Due to its high abundance and geographic dispersion, it is considered an important invasive species, quickly expanding in South America along navigable waterways [1,2]. *Limnoperna fortunei* has a high acclimatization capacity and aggregation behavior on hard surfaces, generating macrofouling that causes environmental and economic issues [3].

Hydroelectric power plants have often been affected by the macrofouling process of the golden mussel. Alterations such as obstructions in pipes and clogging of water collection pumps cause high-cost system shutdowns [4]. To reduce the infestation in industrial facilities, several population control measures are used. Industry used to prefer chemical control options because treatment can be applied throughout the entire facility from a single dosing point [5]. Usually classified as oxidizing and non-oxidizing biocides, these chemical compounds are commonly used in an attempt to control biofouling [6]. Several compounds have been reported as biocides, including chlorine-based compounds [7] and quaternary ammonia compounds [8].

Sodium dichloroisocyanurate (NaDCC) is an organic chlorine-based oxidant, widely used in the control of freshwater mollusks due to its relatively low cost and ease of handling [7]. Studies with *L. fortunei* [7], *Dreissena polymorpha* (Pallas, 1771) [9] and *Perna viridis* (Linnaeus, 1758) [10] show the effectiveness of chlorination on mortality when it is done in low concentrations. The MXD-100™ is a Brazilian commercial non-oxidizing antifouling biocide with active compounds based on extracts of tannins and quaternary ammonia [11]. These substances were developed as control agents for bacteria and algae, and their use extended to mollusks (molluscicides). Efficient results of Bulab 6002™ (a quaternary ammonium polymer similar to MXD-100™) and MXD-100™ were seen in larvae [12] and adults [8] of *L. fortunei*. Understanding the effectiveness of these biocides on mussel morphophysiology is essential as an important part of combating biofouling.

The gills of bivalves have a large surface area for absorption of substances and are in direct contact with the aquatic environment and, consequently, with compounds dissolved in it [13]. As part of the Bivalvia class, *L. fortunei* has two valves articulated around the body, a single foot involving the visceral mass, and two pairs of gills (Ctenidae) [14]. A recent work of our group characterized ultrastructurally the epithelium of the *L. fortunei* gills, showing a high organization of these structures [15]. On the surface, the epithelium is composed by ciliated cells, non-ciliated absorptive cells and mucus-producing cells [16]. Mucus facilitates the capture and transport of particles in the branchial filaments [17]. In this way, gills fulfill the roles of respiration, particle capture and particle transport [18].

Observations of morphological and molecular alterations play an important role in understanding the mechanism of action of population control methods for this species because they measure the response of short and long terms of exposure adjusting control mechanisms [19]. Genes related to environmental stress regulation such as superoxide dismutase (*SOD*) and catalase (*CAT*) act in the rapid elimination of excessive reactive oxygen species (ROS) for organism survival [20]; cytochrome P450 (*CYP*) converts hydrophobic lipid-soluble organic compounds to water-soluble excretable metabolites [21]; and heat shock protein 70 (*HSP70*) protects cells against harmful stress conditions [22]. These have been reported as cellular defense biomarkers in *L. fortunei*. Thus, the exposure of these organisms to stressors can affect the pattern of these gene expressions [23].

In view of the impacts caused by the invasion of the golden mussel and the lack of precise information about the mechanisms of action of biocides, this work aimed to understand the metabolic responses of the *L. fortunei* gills when exposed to the chemical compounds NaDCC and MXD-100™ in different periods, using morphological and molecular analysis. This data will allow us to determine, from pre-established concentrations, the necessary time of exposure for each biocide to be efficient in controlling biofouling.

## 2. Materials and Methods

### 2.1. Mussel Management

Adult specimens of *L. fortunei* were collected from colonies adhered to natural substrates in the reservoir of Volta Grande (VG), Minas Gerais, Brazil (S 20° 05′ 33″; W 48° 06′ 18″). They were packed in cloth bags and transported to the laboratory at the *Centro de Bioengenharia de Espécies Invasoras de Hidrelétricas* (CBEIH). Nearly 170 animals were placed in 36-Liter aquariums containing artesian water (pH 7.7, dissolved oxygen 6.8 mg L^−1^ and turbidity (NTU) 1.68) and maintained at constant aeration and temperatures of 18 to 20 °C. The aquariums contained filters and the water was partially renewed 3 times a week. The animals were fed every two days, with a solution of *Chlorella* and *Spirulina* dissolved in dechlorinated water, about 200 mL per aquarium. After 48 h of acclimation, the mollusks were kept under the same conditions at a temperature of 22 ± 1 °C until the day of the assays. The authorization for activities with scientific purpose is registered in the SISBIO (Sistema de Autorização e Informação em Biodiversidade) under the access code 72222–2.

### 2.2. Exposure to Biocides

Groups of 7 individuals, each approximately 1.5 cm in length, were randomly distributed into 24 glass jars, each one containing 1 L of dechlorinated water under constant aeration. Two short-term and independent bioassays were conducted evaluating the compounds sodium dichloroisocyanurate 60% (Hidrodomi do Brasil Indústria de Domissaneantes LTDA) (NaDCC) (bioassay 1) and MXD-100™ (Maxclean Ambiental e Química SA, MG, Brazil) (bioassay 2) (Figure 1). Each bioassay consisted of three groups with three different exposure times (24, 48 and 72 h) and a dechlorinated-water control. The animals in bioassay 1 were exposed to NaDCC (1.5 mg L^−1^ free available chlorine) during two hours per day, and those in bioassay 2 were exposed to MXD-100™ (0.56 mg L^−1^) for 10 min every 8 h, totaling 30 min daily. The exposure times and concentrations were chosen according to normative instructions for use issued by the Instituto Brasileiro do Meio Ambiente e dos Recursos Naturais Renováveis (IBAMA) [24,25]. These are exclusively approved for use, and frequently used, in treatments in hydroelectric plants. After exposure, the animals were collected, euthanized and preserved. Water quality parameters (temperature, pH, oxygenation, conductivity) were measured twice a day during the experiment (Table 1).

### 2.3. Histological Analysis

Gills from three animals from each glass jar (nine individuals per exposed group and three per control group) were immersed in Bouin’s fixative solution for 24 h. Afterward, they were dehydrated in a progressive series of ethanol, cleared in xylene, and then embedded in paraffin. The transverse histological sections (5 μm) were dewaxed in xylene, hydrated in graded ethanol, and later stained accordingly: sections were stained with hematoxylin-eosin method for structural analysis and with the periodic acid-Schiff (PAS) method combined with Alcian Blue (AB) in pH 2.5 for detection of mucins. Digital image capture was performed with a DM4500 microscope (Leica Microsystems, Wetzlar, Germany).

### 2.4. Morphometry

The epithelium area of the gill was calculated by the difference between the measurements of the internal (delimitation of the haemolymph vessel) and external (contour of the surface of the filament) circumferences of the transverse histological sections of 50 gill filaments per individual (nine individuals per exposed group and three per control group). The morphometric analyses were performed using the software ImageJ (Image-Pro Plus 7.01; National Institutes of Health, Bethesda, MD, USA).

### 2.5. Mucous Cell Analysis

To monitor the effect of exposure to the biocides on the amount of polysaccharides present in cells of the gill epithelium, the number of mucous cells in 50 gill filaments from each individual was counted using ImageJ, as described by David et al. [26].

### 2.6. Scanning Electron Microscopy (SEM) Preparations

For electron microscopy analysis, one golden mussel per glass jar was used (three exposed individuals and one control per treatment). The gills of each of these mussels were dissected, and each gill was observed with a different electron microscopy technique (transmission and scanning). For the observation of surface structures, fragments of the gill were placed into phosphate buffer solution (PBS), then placed in solutions of osmium tetroxide, tannic acid, phosphate buffer and again in osmium tetroxide, respectively. After washing in distilled water, the samples were dehydrated and dried with CO_2_. Finally, the specimens were coated with gold for morphological analysis under a Quanta 200 scanning microscope (FEI Company, Hillsboro, OE, USA). The preparation of samples and their analysis were performed at the Centro de Microscopia at the Universidade Federal de Minas Gerais (UFMG).

### 2.7. Transmission Electron Microscopy (TEM) Preparations

The gill fragments were placed into phosphate buffer solution (PBS), that was then replaced by an appropriate volume of 2% OsO_4_ in PBS (pH 7.3) and incubated in darkness for 2 h at room temperature. After washing samples with distilled water, they were placed in 2% uranyl acetate overnight and kept in darkness. Next, they were washed again and then dehydrated in a series of ethanol bath solutions, followed by an acetone bath series before resin embedding.

The embedding was performed with resin and, after sectioning into 60 nm slices, the tissues were contrasted with uranyl acetate and lead citrate. The preparation of samples and their analysis were performed at the Centro de Microscopia at the Universidade Federal de Minas Gerais (UFMG), using a Tecnai Spirit BioTwin (FEI Company).

### 2.8. Molecular Analysis

#### 2.8.1. Primer Design

We used the available reference genome and gene annotation of *L. fortunei* (PRJNA330677) to recover the nucleotide sequence of the defense genes described by Uliano-Silva et al. [23]. Firstly, to identify the genes of interest among the predicted genes, we performed a similarity search using BLASTP against the UniProt database. Then, we searched for the product name of our genes of interest in the BLASTP results, only selecting hits with over 70% identity and E-value lower than 0.001. To further confirm the identity of the selected genes, we performed an additional similarity search with BLASTP, but against the NCBI non-redundant protein database, only retaining genes for which known homologs were found in other mussel species. Additionally, we also mapped the defense gene transcripts reported by Uliano-Silva et al. [23] into the genome with Minimap2 (using the splice preset). Then, we retained only mapped transcripts with a MAPQ equal to 60 that overlapped the genome coordinates of predicted genes present in the gene annotation. Again, to confirm that the identity of the overlapped predicted genes matched the genes reported by Uliano-Silva et al. [23], we performed a blastp search against the NCBI non-redundant protein database, considering only genes related to the transcriptome study or other mussel species. Finally, with the genome coordinates of all the genes of interest (including GAPDH as a constitutive gene), we recovered the nucleotide sequence of the CDS for the predicted transcript of each gene using the getfasta function in the BEDtools software suite (with the -split and -s parameters). Those nucleotide sequences were then used to design primers in the software Primer3Plus https://www.bioinformatics.nl/cgi-bin/primer3plus/primer3plus.cgi (accessed on 14 September 2021). Table 2 shows the primer pairs used to evaluate gene expression.

#### 2.8.2. Gene Expression

Three animals per glass jar (totaling nine individuals per exposed group and three per control group) were submitted to total RNA extraction according to instructions from the TRI Reagent^®^ (Sigma, St. Louis, MO, USA) manufacturer. An amount of 1.5 µg of total RNA was converted to cDNA following the manufacturer’s instructions with the RevertAidTM H. Minus First Strand cDNA Synthesis kit (Thermo Scientific, Waltham, MA, USA). The RT-qPCR reaction mixture was prepared by combining 5 µL of iTAqTM Universal SYBR^®^ Green Supermix (Bio-Rad, Hercules, CA, USA), 0.4 µM of each primer and 1 µL of cDNA diluted to a 1:10 ratio in nuclease-free water, and adjusted to a final volume of 10 µL with nuclease-free water. The thermal cycling conditions involved an initial denaturation step at 95 °C for 2 min, followed by 45 cycles of denaturation at 94 °C for 15 s, annealing at 60 °C for 15 s and extension at 72 °C for 20 s. After the amplification, an additional extension step at 72 °C for 5 min was performed. To assess the specific melting temperature of each primer set, a dissociation step was conducted at the end of the amplification process. Amplicon sequencing was performed on the 3730 DNA Analyzer (Applied Biosystems, Waltham, MA, USA) by a commercial laboratory and the sequences deposited in GenBank (Table 2).

### 2.9. Statistical Analysis

To test how the number of mucous cells and the area of the epithelium were influenced by the presence of MXD-100™ and NaDCC, over time and between groups, generalized linear models (GLMs, following the Crawley, [27] protocol) were built using the R platform (R Development Core Team 2015). Gene expression analysis was conducted using REST software [28], employing a pairwise fixed reallocation randomization test and taking into account primer efficiency. The data are presented as means ± SEM (standard error of the mean), with statistical significance defined as a *p*-value < 0.05. All analyses were carried out using GraphPad Prism (version 8.0, San Diego, CA, USA).

## 3. Results

### 3.1. Changes from Exposure to Sodium Dichloroisocyanurate (NaDCC)

No morphological changes were observed in the gill filaments in individuals in the control group (Figure 2A–C); however, individuals exposed to NaDCC showed progressive filament degradation and ciliary deformation, from mild to severe, during the three exposure periods—24 h (Figure 2D–F), 48 h (Figure 2G–I), and 72 h (Figure 2J–L). The cilia were loose in the non-ciliated region of the filaments.

The increase in area of the epithelium was observed between the groups (*p* < 0.001, F = 415.42, Df = 1), and in the period of 24 h, in relation to 48 h (*p* < 0.001, F = 172.18, Df = 2) (Figure 3). In addition, when compared to the control group (Figure 4A), histological sections showed lifting of the epithelium during the periods of 24 and 48 h of exposure (Figure 4B,C) and epithelial desquamation from the second exposure period of 48 h (Figure 4C,D).

There was an increase in the number of mucus-producing cells in animals exposed to NaDCC for 24 h (*p* < 0.001, residual deviance = 114.5, Df = 1) (Figure 5A), which was not observed in the group exposed to 48 h (Figure 5B,C). After 72 h of exposure to NaDCC, the number of cells could not be counted due to the mortality of animals in this period. It was also possible to observe a large accumulation of mucus on the front surface of the filaments in individuals exposed for all periods (Figure 2E,H,K).

As for the ultra-morphological aspect, during the exposures to all treatment times, gill epithelial cells showed autophagosomes in the cytoplasm (Figure 6B,C). The exposure for 48 and 72 h showed cytoplasmic dilatations (Figure 6C,D). The nuclei of the cells had clusters of chromatin and a clear appearance only in the 72-h period (Figure 6D).

Treatment with NaDCC was able to upregulate *SOD* (~20.7-fold change) and *HSP70* (26.8-fold change) expressions during the 24-h exposure, compared to the control (Figure 7A). There was no statistical difference at 48 (Figure 7B) and 72 h (Figure 7C) of exposure for these transcripts. The *CYP* gene did not change its expression in any of the NaDCC exposure treatments, and *CAT* did not amplify during the 72-h exposure (Figure 7C).

### 3.2. Changes from Exposure to MXD-100™

Individuals exposed to MXD-100™ presented filaments in a severe stage of deformation since the first period of exposure (24 h) (Figure 8D,G,J), while those in the control group showed no morphological changes (Figure 8A–C). Severe ciliary loss was also identified from the shortest time of exposure (Figure 8E,H,K). There was a significant effect on the area of the epithelium between groups (*p* < 0.001, F = 77.59, Df = 1) and between time periods (*p* < 0.001, F = 107.21, Df = 1). The increase in area of the epithelium was observed in the group exposed for 24 h, but epithelium area decreased in animals exposed for 48 h. It was not possible to observe it in the group exposed to MXD-100™ for 72 h because all individuals died (Figure 9).

In histological sections, a moderate to severe lifting of the epithelium was observed after 24 and 48 h of exposure (Figure 10B,C). In relation to epithelial desquamation, it was moderate after the first exposure time (24 h), and severe after 72 h (Figure 10B–D). A hyperplastic increase could be seen during all three exposure periods (24, 48, 72 h) to MXD-100™, mainly in epithelial cells in the frontal region of the filaments (Figure 10B–D). No morphological alterations could be observed in the control group (Figure 10A).

An accumulation of mucus on the front surface of the filaments was also identified in individuals in all periods of exposure to MXD-100™ (Figure 8F,L), but no statistical difference in the number of mucus-producing epithelial cells could be observed between the 24-h and 48-h groups. This number could not be assessed at the 72-h group as the cells were no longer present.

The internal cellular organization of the groups exposed to MXD-100™ for 24 and 48 h showed translucent cytoplasm with dilations, lamellar bodies and few organelles (Figure 11B,C). The nuclei had a clear appearance with a chromatin cluster, and in the 48-h group, exclusively, the nuclear envelope was damaged (Figure 11C). In animals exposed to MXD-100™ for 72 h, the cells had become scarcer and looser, with indistinguishable cytoplasm and a high degree of degeneration (Figure 11D).

Changes in gene expression were observed in animals exposed to MXD-100™ even in the shortest period of exposure. *CYP*, *SOD*, *CAT*, and *HSP70* were upregulated during the 24-h period, compared to the control (Figure 12A). *CAT* expression was downregulated during the 48-h period of exposure, compared to the control (Figure 12B). The 72-h period of exposure to MXD-100™ did not present any changes in the measured markers (Figure 12C).

## 4. Discussion

The gills are one of the first target organs for several xenobiotic compounds due to their direct contact with the contaminants in water [29]. In this study, the gills of *L. fortunei* were evaluated after exposure for different time periods to two commonly used biocides.

Morphological knowledge provides information about the health status of an organism and an understanding of the action of population control methods [30]. Our results showed constant changes in the gill epithelium in response to both treatments (NaDCC and MXD-100™). This reinforces their vulnerability [31], in addition to emphasizing how phenotypic plasticity is an important mechanism for adaptation to shifts in environmental conditions [32].

Furthermore, differences in gene expression can provide information on specific cellular responses in mollusks [33]. Several biochemical markers are used to evaluate toxicity in bivalves. These biomarkers include parameters related to the oxidation of biomolecules, changes in the levels of antioxidants, detoxification and general metabolic enzymes [34]. For example, Girardelo et al. [35] observed a decrease in *SOD* levels in golden mussels after exposure to zinc oxide nanoparticles, modulating its antioxidant defenses in response to oxidative stress, and Wang et al. [36] observed the detoxification and metabolism of tetrabromobisfenol A, an environmental contaminant, from alterations in the expression of biotransformation genes, such as *CYP* in Mediterranean mussel (*Mytilus galloprovincialisalis*).

NaDCC has a molluscicidal action based on the oxidation of organic matter, which confers toxic and lethal effects on the target organisms [37]. When in contact with water, NaDCC releases free available chlorine in the form of hypochlorous acid (HOCl) [38]. Beyond that, isocyanurates compounds are derived from cyanuric acid, the carrier that allows the chlorine to be contained in a solid, stable and dry form [38]. It has been observed that chlorine induces oxidative stress in cells by generating reactive oxygen species (ROS), which can damage cell membranes, proteins and nucleic acids [39]. Consistent with this, our results showed that exposure to NaDCC for 24 h led to an upregulation of the expression of superoxide dismutase (*SOD*), compared to the control. *SOD*s are universal enzymes of organisms that live in the presence of oxygen [40]. They decrease ROS levels in oxidative stress by reducing the superoxide ion radical to hydrogen peroxide [41]. In addition, during the 24 h of exposure there was also an upregulation of heat shock protein 70 (*HSP70*) expression, compared to the control. *HSP70* acts in cellular protection against harmful conditions by binding and refolding damaged proteins, playing an important role in the response to cellular stress [22]. These results possibly show that NaDCC induces an acute cellular response to the oxidative stress it generates.

In the same period of 24 h of exposure, the epithelium area grew and the number of mucous cells was higher, probably because the gills respond to the first contact to the biocide, trying to avoid the entry of substances into the tissue through passive diffusion and forming inert compounds that can be excreted as pseudofeces [42]. This excretion is due to the capacity of bivalves to eliminate rejected material, usually inorganic fractions [43,44].

During the 72 h of exposure to NaDCC, no expression of the *CAT* gene was observed. *SOD* and *CAT* are antioxidant enzymes that operate in the same pathway, acting in the defense against oxidative stress by reactive oxygen species (ROS). While *SOD* accelerates the reaction between the superoxide and hydrogen peroxide, *CAT* eliminates hydrogen peroxide in cells [45]. Thus, the *SOD/CAT* pathway may have been blocked in 72 h of exposure, stopping the expression of the *CAT* gene.

Other morphological changes occurred gradually during the three exposure periods to NaDCC. The ciliary damage possibly occurred by direct contact with the chemical [46], and lifting and desquamation of the epithelium occurred as a form of gill defense mechanisms against frequent exposure to toxic agents [47].

Cytoplasmic and cellular damage appeared gradually during NaDCC exposures, including autophagolysosomes that can also be observed in the three periods, being a conserved cytoprotective mechanism activated by environmental stimuli [48]. Thus, from the changes related to protection mechanisms observed here—molecular (increased *SOD* and *HSP70*), morphological (increased mucus production and epithelium area), and ultramorphological (presence of autophagosomes) changes—it seems that the alterations may be reversible up to the 48 h of exposure to NaDCC, while only in the 72-h period there were serious cellular alterations related to cell death processes (epithelial desquamation, nuclei with chromatin compaction and a clear appearance). In other words, in 72 h, the cellular adaptive response can no longer accommodate the stressor [48]. These data are consistent with previous studies that observed that chlorination at low doses is effective for mussel elimination, but only when it occurs continuously. When the treatment is applied only for a short period, mussels can physiologically recover from it in a process of cellular regeneration after recent exposure to toxicants [10,39,49]. This probably happens because the animals close their shells when in contact with the biocide, delaying the effect of the compound.

Regarding MXD-100™, it is reported to be relatively inert to the internal infrastructure of industrial water systems, effective in low concentrations, quickly inactivated and easy to handle [50]. It is composed of a combination of tannins and quaternary ammonium (QAC) that bind to the negatively charged surface of mollusk membranes, causing membrane rupture and leakage of the intracellular constituents [6,51]. Compatible with this information, a study has shown that tannin formulations have acute toxicity for golden mussels [52].

In our experiments, the animals exposed to MXD-100™ for 24 h have shown an upregulation of the expression of *SOD*, *CAT*, *CYP* and *HSP70* gene markers, compared to the ones in the control group. The affected cells have antioxidant systems which limit the effects of the ROS, and these systems are composed of molecules and enzymes such as *SOD* and *CAT* [53]. From our observations, the upregulation of the expression of these genes plus the upregulation of *HSP70* have shown a cellular stress response more intense and genetically diverse than those observed after NaDCC exposures. Further, an upregulation in *CYP* expression was observed after 24 h of exposure to MXD-100™, given that the cytochrome P450 enzymes (*CYP*) are responsible for the oxidative metabolism of endogenous and xenobiotic compounds, playing a significant role in the biosynthesis of endogenous compounds and the biotransformation of xenobiotics [54]. Thus, in addition to the cellular response to oxidative stress, there was also activation of xenobiotic biotransformation and detoxification processes in an attempt to eliminate MXD-100™ from the mussel gills.

There was no change in *SOD* gene expression levels after 48 h of exposure to MXD-100™. There was, however, a downregulation of the expression of *CAT* transcripts, compared to the control. This downregulation probably occurred because there was no action of *SOD* on the transformation of superoxide into hydrogen peroxide to be eliminated by *CAT*. In other words, 24 h of exposure may increase enzymatic activity, but 48 h of exposure can inhibit enzymatic activity caused by oxidative stress, reducing defense mechanisms [55].

Cell damage related to cell death was observed during exposure to MXD-100™, such as translucent cytoplasm, nuclear damage and the presence of lamellar bodies. It may happen because reactive oxygen species are genotoxic [56] and can activate apoptosis pathways [42,48]. Apoptosis is a type of programmed cell death that produces changes in cell morphology and biochemical processes, and may occur in response to cell damage caused by toxic agents [57]. The lamellar bodies observed here are formed by layers of concentric membranes produced during the autophagic process [58], being indicative of the activation of the apoptotic pathway.

The structural architecture of the gill epithelium was also altered due to the acute toxicity of MXD-100™. The increase in epithelium area was seen only for 24 h as a means of protection, trying to isolate the organism from the environment [59]. However, it is possible that the number of mucous cells increased before 24 h due to the high toxicity of the compound. The decrease in the area of the epithelium in 48 h might have occurred due to the high cytotoxicity of MXD-100™, which led to rapid cell death. Other changes (desquamation, epithelial lifting, hyperplasia) remained severe at all times, as a way of trying to decrease the respiratory surface to inhibit the absorption of compounds and increase in diffusion distance [60]. These acute changes caused by MXD-100™ from 24 h of exposure may have occurred because the mussel cannot detect this biocide as a harmful compound and the closing response is not provoked [61].

Changes in the environment are able to trigger changes in the cellular response; thus, the timing and success of these changes ultimately determine whether the cell survives and acclimates, or if it dies [62]. According to the results obtained in this study, NaDCC (1.5 mg L^−1^) shows lethal damage after 72 h of exposure, with gill cells showing reversible changes up to 48 h, while MXD-100™ (0.56 mg L^−1^) shows severe and irreversible gill changes after 24 h of exposure. Thereby, ultramorphological and molecular analyses of gills might be efficient in identifying the toxicity of chemical biocides in golden mussels.

## 5. Conclusions

The search for the “ideal biocide” to effectively combat the invasion of *L. Fortunei* into aquatic systems while minimizing environmental impact remains ongoing. Despite efforts, a suitable antifouling compound has yet to be discovered, necessitating investigations focused on reducing its environmental footprint [63]. Our data analysis enabled the determination of the required exposure time to efficient biocides for biofouling control at pre-established concentrations. The results indicated that MXD-100™ exhibited acute toxicity in a shorter period of time (24 h), while NaDCC potentially requires continuous exposure for species control. Discrepancies between these findings and existing literature may arise from the use of static bioassays, emphasizing the need for further studies assessing dose–response effects on gills and conducting prolonged action studies to account for chronic effects. Beyond gill evaluation, a comprehensive assessment of other tissues in *L. Fortunei* is essential to evaluate the overall effectiveness of biocide treatments.

## Figures and Tables

**Figure 1 animals-13-03258-f001:**
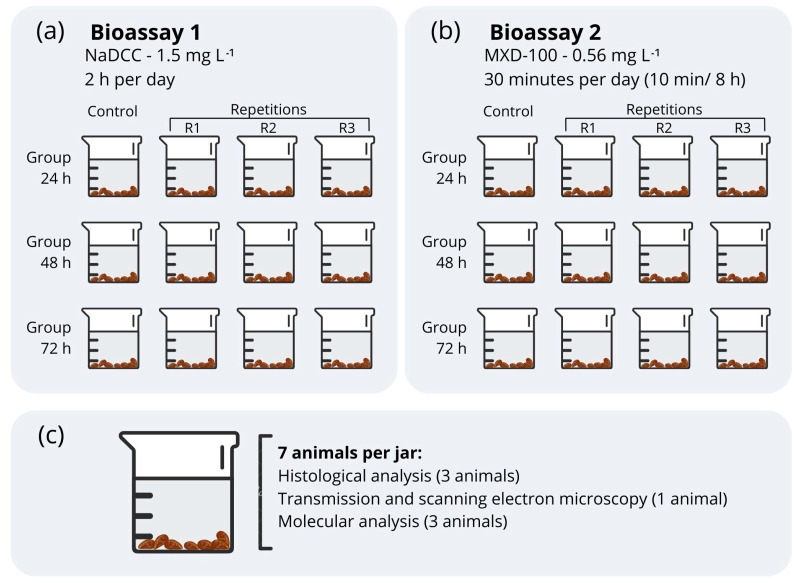
Scheme representing the methodology of bioassay 1 (sodium dichloroisocyanurate) (**a**) and bioassay 2 (MXD-100™) (**b**). (**c**) Number of animals in jar per analysis.

**Figure 2 animals-13-03258-f002:**
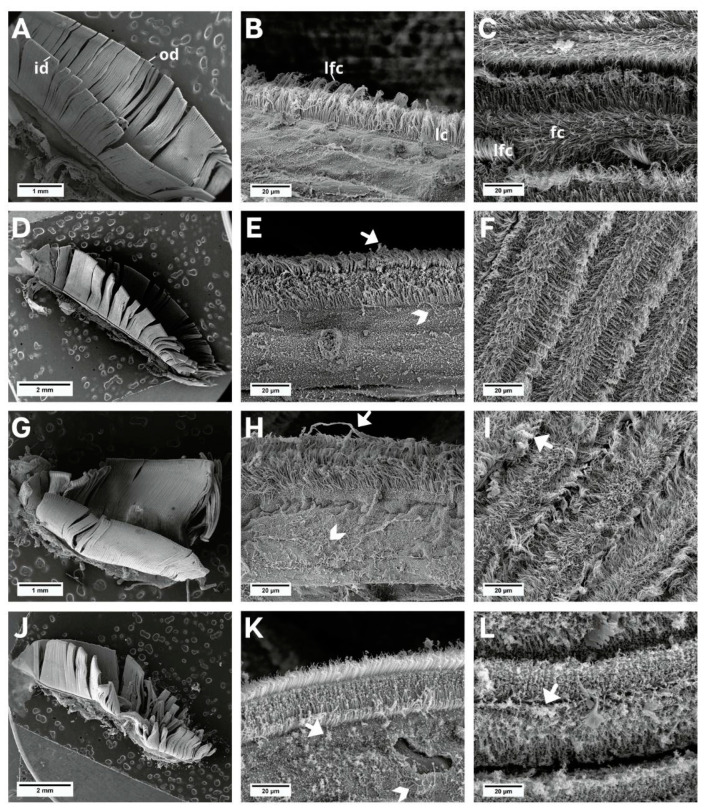
SEM images of the *L. fortunei gills*. (**A**–**C**) Control group and (**D**–**L**) groups exposed to NaDCC. (**A**) Ciliated gill with inner (id) and outer demibranch (od). (**B**) Dorsal view of gill lamellae showing the frontal region of the filaments with lateral cilia (lc) and laterofrontal cirri (lfc). (**C**) View in perspective of a set of filaments, showing frontal cilia (fc) and laterofrontal cirri (lfc). (**D**–**F**) Group 24 h after conditional exposure. (**G**–**I**) Group 48 h after conditional exposure. (**J**–**L**) Group 72 h after conditional exposure (presumably they were dead, as their valves were open). Mucus (white arrow); loose cilia (arrow head).

**Figure 3 animals-13-03258-f003:**
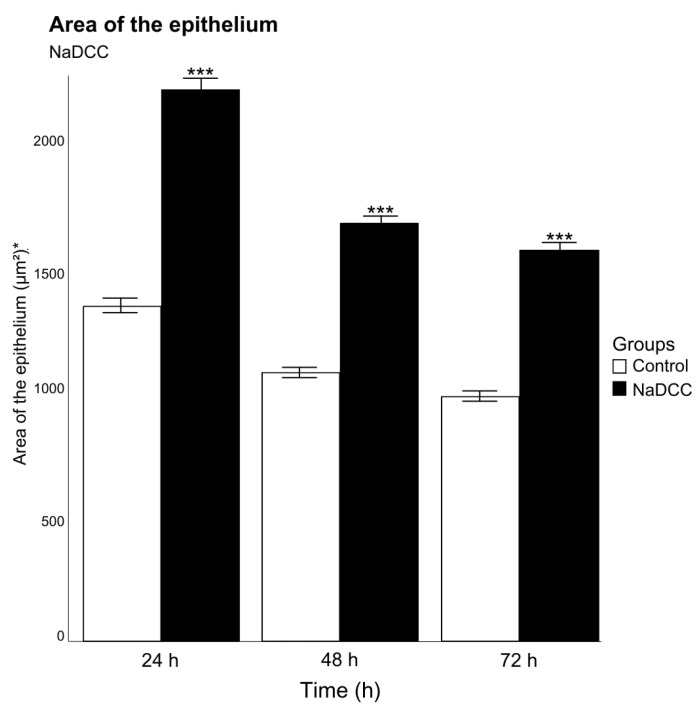
Mean and standard deviation of epithelium area in 50 branchial filaments exposed to sodium dichloroisocyanurate for different periods (24, 48 and 72 h after conditional exposure). * Measurement unit: square micrometer. Significant differences between exposed groups are represented by *** *p* < 0.01.

**Figure 4 animals-13-03258-f004:**
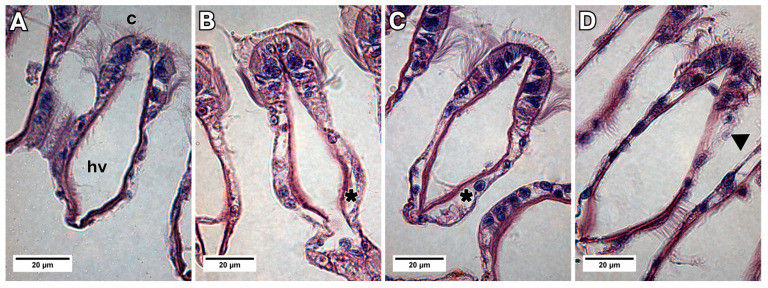
Light micrograph images showing the transverse sections of a gill filament of *L. fortunei* exposed to NaDCC. (**A**) Control group. (**B**) Group 24 h after conditional exposure. (**C**) Group 48 h after conditional exposure. (**D**) Group 72 h after conditional exposure (presumably they were dead, as their valves were open). Cilia (c), haemolymph vessel (hv), epithelium lifting (*) and epithelial desquamation (arrow head).

**Figure 5 animals-13-03258-f005:**
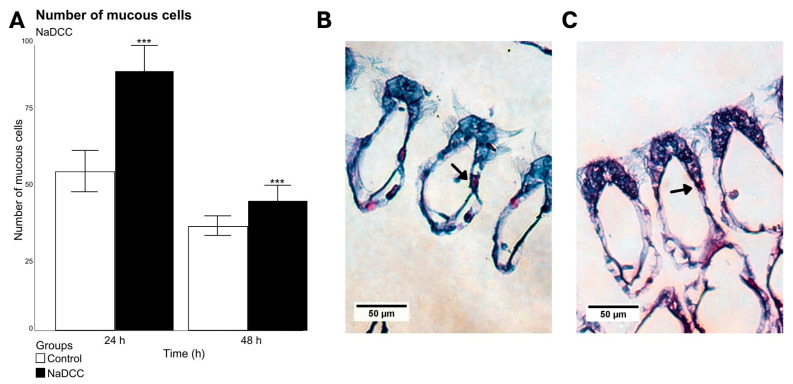
Mean and standard deviation of the number of mucous cells present in 50 branchial filaments to NaDCC exposed (24 and 48 h after conditional exposure) (**A**). Significant differences between exposure groups are represented by *** *p* < 0.01. Group exposed for 72 h presented a mortality rate of almost 100%, as their valves were open. (**B**,**C**) Light micrograph images showing the transverse sections of a gill filament of *L. fortunei* stained with AB-PAS. (**B**) Group 24 h after conditional exposure. (**C**) Group 48 h after conditional exposure. Mucus-producing cells (arrow).

**Figure 6 animals-13-03258-f006:**
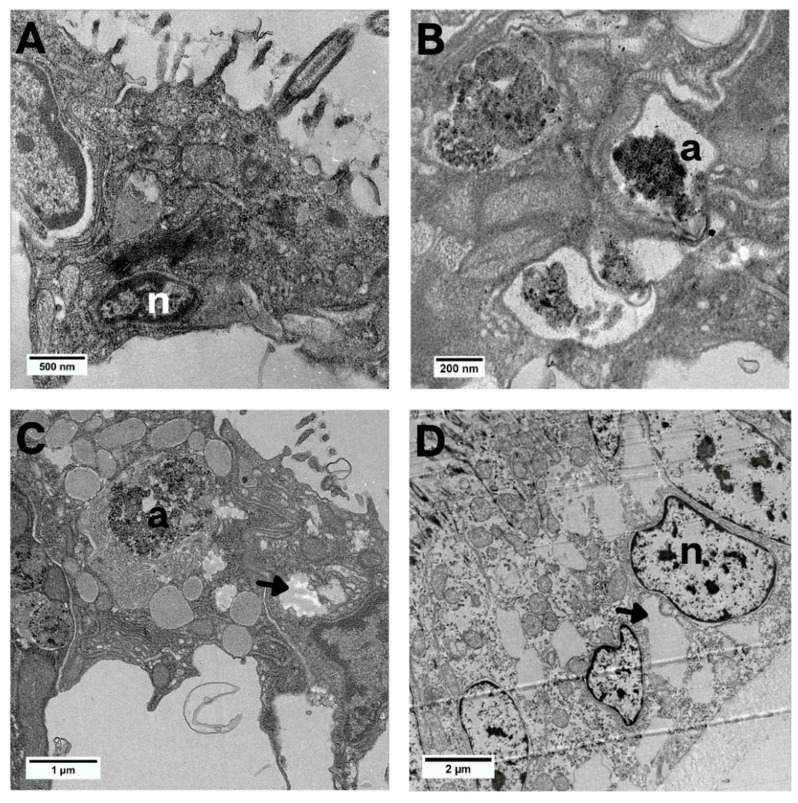
Bright-field TEM images of the epithelial cells of *L. fortunei* exposed to NaDCC. (**A**) Control group. (**B**) Group 24 h after conditional exposure. (**C**) Group 48 h after conditional exposure. (**D**) Group 72 h after conditional exposure (presumably they were dead, as their valves were open). Autophagosome (a), nuclei (n) and cytoplasmic dilatations (arrow head).

**Figure 7 animals-13-03258-f007:**
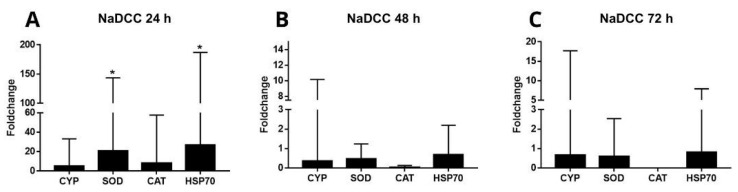
Expression profiles of transcripts associated with defensive enzymes *CYP*, *SOD*, *CAT*, and *HSP70* in the gills of *L. fortunei* exposed to NaDCC compared to a control group maintained in dechlorinated water with continuous aeration. Three treatment durations were investigated: (**A**) 24 h after conditional exposure, (**B**) 48 h after conditional exposure, and (**C**) 72 h after conditional exposure. Data are represented as mean ± SEM (standard error of the mean). Statistically significant differences at α = 0.05 are denoted by *.

**Figure 8 animals-13-03258-f008:**
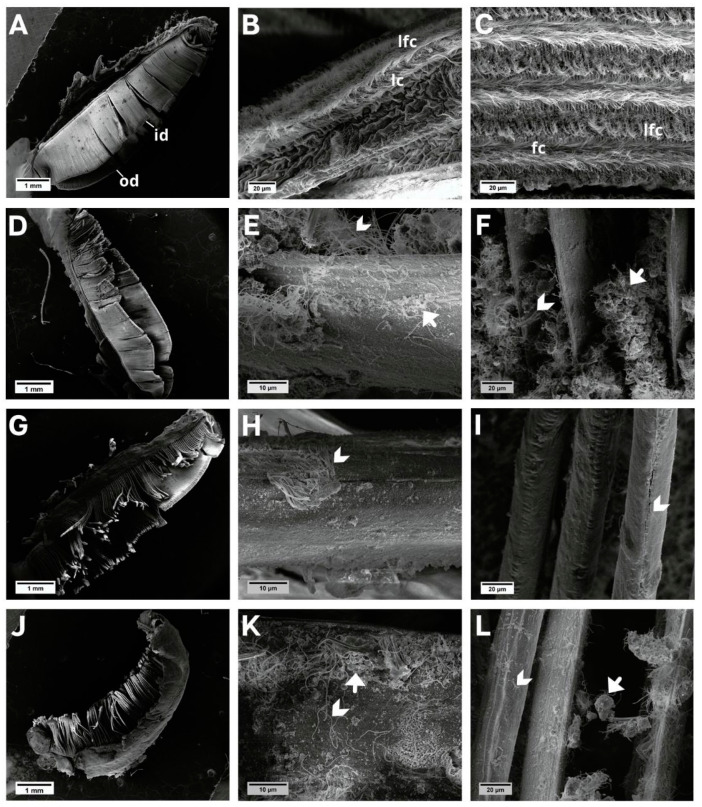
SEM images of the *L. fortunei* gills. (**A**–**C**) Control group and (**D**–**L**) groups exposed to MXD-100™. (**A**) Ciliated gill with inner (id) and outer demibranch (od). (**B**) Dorsal view of gill lamellae showing the frontal region of the filaments, lateral cilia (lc) and laterofrontal cirri (lfc). (**C**) View in perspective of a set of filaments, showing frontal cilia (fc) and laterofrontal cirri (lfc). (**D**–**F**) Group 24 h after conditional exposure. (**G**–**I**) Group 48 h after conditional exposure. (**J**–**L**) Group 72 h after conditional exposure (presumably they were dead, as their valves were open). Mucus (white arrow); loose cilia (arrow head).

**Figure 9 animals-13-03258-f009:**
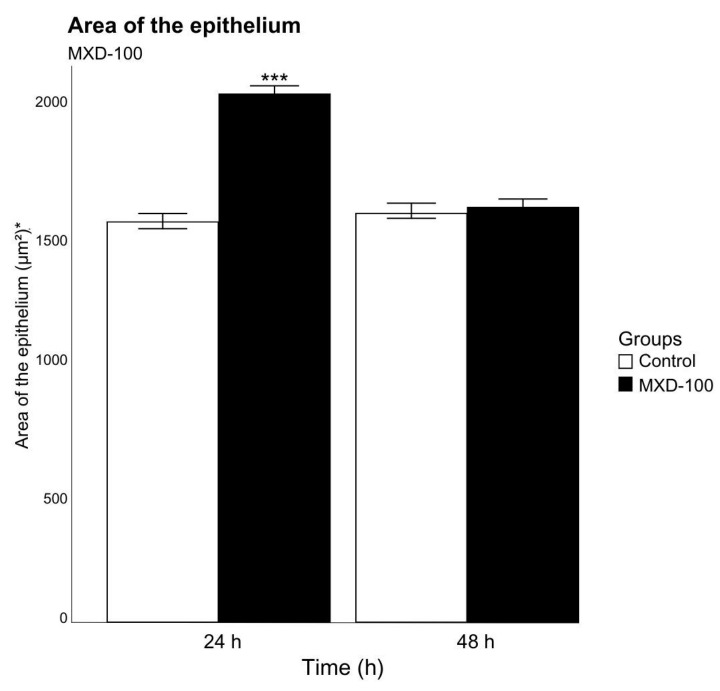
Mean and standard deviation of epithelium area in 50 branchial filaments exposed to MXD-100™ for different periods (24 and 48 h after conditional exposure). All individuals died in the 72-h after conditional exposure. * Measurement unit: square micrometer. Significant differences between exposure groups are represented by *** *p* < 0.01.

**Figure 10 animals-13-03258-f010:**
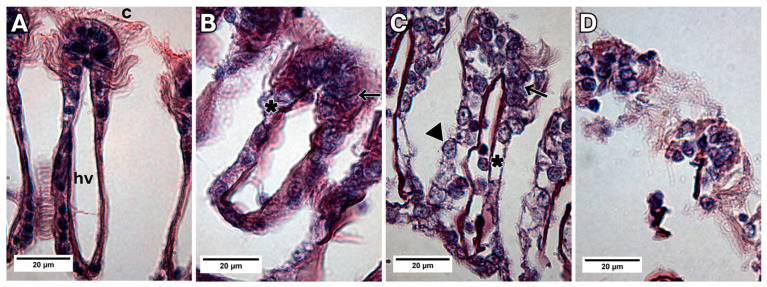
Light micrograph images showing the transverse sections of a gill filament of *L. fortunei* exposed to MXD-100™. (**A**) Control group. (**B**) Group 24 h after conditional exposure. (**C**) Group 48 h after conditional exposure. (**D**) Group 72 h after conditional exposure (presumably they were dead, as their valves were open). Cilia (c), haemolymph vessel (hv), hyperplasia (arrow), epithelial desquamation (arrow head) and epithelium lifting (*).

**Figure 11 animals-13-03258-f011:**
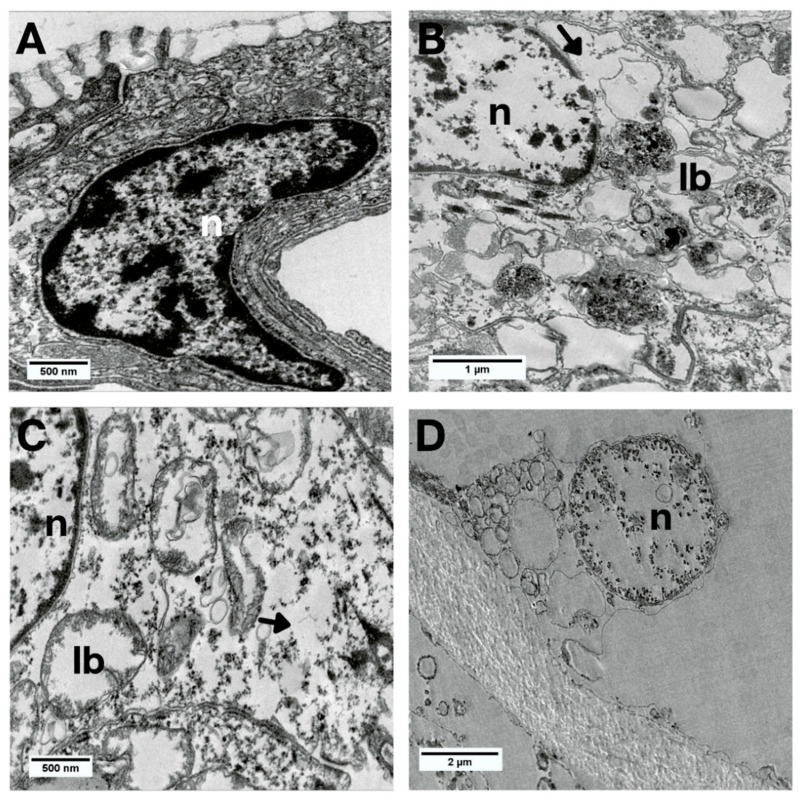
Bright-field TEM images of the epithelial cells of *L. fortunei* with MXD-100™. (**A**) Control group. (**B**) Group 24 h after conditional exposure. (**C**) Group 48 h after conditional exposure. (**D**) Group 72 h after conditional exposure (presumably they were dead, as their valves were open). Lamellar bodies (lb), nuclei (n) and cytoplasmic dilatations (arrow head).

**Figure 12 animals-13-03258-f012:**
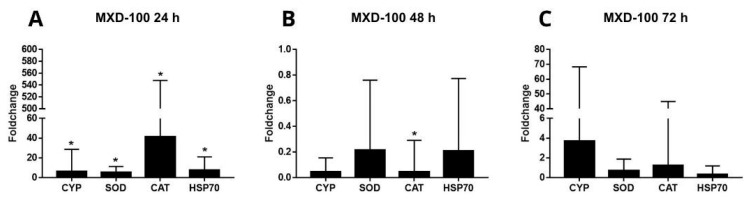
Expression profiles of transcripts associated with defensive enzymes *CYP, SOD, CAT,* and *HSP70* in the gills of *L. fortunei* exposed to MDX-100 compared to a control group maintained in dechlorinated water with continuous aeration. Three treatment durations were investigated: (**A**) (24 h after conditional exposure), (**B**) (48 h after conditional exposure), and (**C**) (72 h after conditional exposure). Data is represented as mean ± SEM (standard error of the mean). Statistically significant differences are denoted by * *p* < 0.05.

**Table 1 animals-13-03258-t001:** Mean of water quality parameters (pH, temperature, oxygenation, conductivity) measured twice a day.

**Water Quality Parameters with NaDCC Biocide (1.5 mg L^−1^)**				
**Treatment**	**pH**	**Temperature (°C)**	**DO * (mg L^−1^)**	**Conductivity (μS/cm)**
Control 24 h (wqi = 2, mean, SD)	7.69 (0.10)	21.90 (0.00)	8.45 (0.52)	134.00 (2.83)
24 h (wqi = 6, mean, SD)	7.68 (0.09)	22.07 (0.14)	8.93 (0.23)	134.50 (1.05)
Control 48 h (wqi = 4, mean, SD)	7.71 (0.16)	21.85 (0.33)	8.95 (0.24)	131.75 (3.40)
48 h (wqi = 12, mean, SD)	7.71 (0.10)	21.78 (0.36)	8.90 (0.13)	133.67 (3.31)
Control 72 h (wqi = 6, mean, SD)	7.69 (0.17)	22.23 (0.43)	9.30 (0.42)	132.17 (3.31)
72 h (wqi = 18, mean, SD)	7.69 (0.10)	22.01 (0.44)	8.71 (0.27)	132.44 (3.35)
**Water Quality Parameters with MXD-100™ Biocide** (**5.6 mg L^−1^**)				
**Treatment**	**pH**	**Temperature** (**°C**)	**DO *** (**mg L^−1^**)	**Conductivity** (**μS/cm**)
Control 24 h (wqi = 2, mean, SD)	8.13 (0.10)	20.50 (0.57)	7.30 (0.85)	151.45 (4.74)
24 h (wqi = 6, mean, SD)	7.75 (0.15)	20.55 (0.36)	7.25 (0.29)	149.80 (1.38)
Control 48 h (wqi = 4, mean, SD)	8.29 (0.38)	20.60 (0.33)	7.07 (0.61)	150.92 (2.80)
48 h (wqi = 12, mean, SD)	7.78 (0.13)	20.50 (0.29)	7.29 (0.46)	150.15 (1.52)
Control 72 h (wqi = 5, mean, SD)	8.16 (0.46)	19.98 (0.18)	7.00 (0.64)	150.24 (1.49)
72 h (wqi = 15, mean, SD)	7.88 (0.36)	20.04 (0.19)	7.10 (0.40)	150.05 (1.45)

wqi = number of times the water quality index was measured; SD = standard deviation; * dissolved oxygen.

**Table 2 animals-13-03258-t002:** Primers used for the qRT-PCR assays. *CYP*: cytochrome P450; *SOD*: superoxide dismutase; *CAT*: catalase; *HSP70*: heat shock protein 70. F: forward primer; R: reverse primer. BP: base pairs. The annealing temperature of all reactions was 60 °C.

Gene	Initial	Sequence	Genbank Access Entries
*Cytochrome P450*	*CYP*	F: 5′-GCCAAGGGACACGATTCTAC-3′ R: 5′-CATACAAAGACACCGCCATTC-3′	OP066521
*Superoxide dismutase*	*SOD*	F: 5′-TGATTCGTAGGGACTTTGGA-3′ R: 5′-CCTGGTTAGCACAAGCAACA-3′	OP066522
*Catalase*	*CAT*	F:5′-CTCTCAAGTCGGTGTATTCTGG-3′ F:5′-CCTGTTTTTCCTTTCGCTGT-3′	OP066523
*Heat Shock Protein 70*	*HSP70*	F:5′-GAAGCCTACCTCGGACAAAA-3′	OP066524
		F:5′-CGGACCTCAAACAGTGAACC-3′	
*Glyceraldehyde 3-phosphate dehydrogenase*	*GAPDH* *	F: 5′-TACTGCGAGGCAACAATAGG-3′	OP066520
		R: 5′-GTTTTTCAAGGCACCACAGA-3′	

* The primer *GAPDH* (Glyceraldehyde 3-phosphate dehydrogenase) was the reference.

## Data Availability

All data needed to evaluate the conclusions in this paper are present in the text.

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
