# Peer review of "Acute Exposure to Two Biocides Causes Morphological and Molecular Changes in the Gill Ciliary Epithelium of the Invasive Golden Mussel Limnoperna fortunei (Dunker, 1857)"

_animals, 2023, doi:10.3390/ani13203258_

Round 1

Reviewer 1 Report

The manuscript is quite clear, relevant for the field and presented in a fairly well-structured manner.

The information provided in the methods section is not always clear enough:

134-136, 140-144: Table 1. It would be necessary to explain what "n" is in  treatment column and why the sample is different (n =  from 2 to 18), if before the text states that the number of samples is the same everywhere (Each bioassay consisted of three groups with different ex-125 pose times each (24, 48 and 72 hours), plus a control group kept in dechlorinated water 126 only. All assays were conducted in triplicates.) Another suggestion - Measurements are averaged, so the title (Table 1) does not match the content. What's more - the additional names (in 142 and 143) are formulated also inaccurate - water quality parameters appear to be like NaDCC or MXD-100. 

155-158. Section 2.4. It is not very clear how those measurements are arranged on the transversal histological sections in order to maintain the uniformity of the measurement.

228-232. Section 2.9. Too narrow an explanation of where, why and how tests were applied. General information provided with general reference. It would be good to clarify methods and their suitability for specific tasks.

Visualisation of results - insuficient. The quality of the photos is excellent, but the design of the graphics should be unified (color graphics in one place, black-and-white graphics in others look inappropriately). In Fig. 7 (291-293) and Fig. 12 (349-351) control columns are required. The statistical reliability test should be specified in the description parts: 285-290 and 343-348.

In the discussion part, it would be worth discussing the issue of possible gill regeneration.

Recommendation - do not mix grouping and explanatory letter forms (uppercase and lowercase) when describing photo images in both text and titles. If capital letters are used in the text, they must also be capitalized in the images; respectively with grouping.

In addition to the letters in the titles, there are other proofreading errors (incorrect spaces in mathematical notes, dashes, comma in the middle of a number, authors' names in both upper and lower case letters in the literature source). Advice to follow the requirements of Zoological nomenclature not only in the text, but also in the literature list (specifically, Latin species names should be italic).

The conclusion consistent with the evidence and arguments presented.

I have no further comments, the work is interesting, multifaceted and has practical significance.

Reviewer 2 Report

The manuscript titled “Acute exposure to two biocides causes morphological and molecular changes in the gill ciliary epithelium of the invasive golden mussel Limnoperna fortunei (Dunker, 1857)shows the effects of the biocides - MXD-100 and NaDCC - on the bivalve gills by morphological analyses with different techniques and gene expression related to defense system.

The research is a well done descriptive work, very interested for this journal special Issue. Materials and methods are well developed and the results show which biocide had acute toxicity in a shorter time, in concordance with the conclusions.  The paper is well structured, interesting and easy to read and the statistical analysis well applied. I recommended two minor revisions.

The authors proposed to investigate the effects of different periods (24, 48 and 72 h) of exposure to the known biocides MXD-100 (0.56 mg/L) and NaDCC (1.5 mg/L), on the gills of the invasive Limnoperna fortunei through morphological and molecular analyses.

I consider the topic appropriate and interesting as this special Issue is about Invasions, Alien and Pest Molluscs. The authors well mentioned a different way to do the bioassays for future researches, with continue flow instead of statics to get a better comparison with other authors. 

The paper is descriptive, with two known biocides, but the authors got good results with the comparison between them and useful to carry out in a more practical way.

The methodology is ok for my understanding and the controls are well done, the authors could try next time to do the bioassays in a way closer to the reality, could be assays “in situ”.

The conclusions are consistence with the results obtained by the authors. They compared two biocides, with known concentrations, and obtained good results under control lab conditions. They concluded which biocide had acute toxicity in a shorter time and therefore which is most useful, cheap and less contaminant. Besides, they proposed morphological and gene expression analyses as biomarkers.

The references are relevant and appropriate and the tables and figures are clear, only minor comments:

Page 9 Line 260, Fig 4. Please clarify “LM” as Light Micrograph images as was not described before

Page 13.  Line 323, Fig 10. It is hard to see the reference letters in the LM images. Please change the color

Reviewer 3 Report

1.In the abstract section. The authors only described the change in gene expression (SOD, CAT, HSP70, and CYP450) in MXD-100-exposed L. frtunei, but not NaDCC. This will lead the audience to misunderstand that the authors only examined gene expression in MXD-100-exposed L. frtunei. I would suggest that the authors reorganize the abstract to have a concise summary. This will make it easier for the audience to understand their study.

2. Line 131-132. "...by the Instituto Brasileiro do Meio Ambiente e dos Recursos Naturais Renováveis (IBAMA), nº 171 and nº 182, of October 21, 2015". It is better to have the citation and show the website link in the reference list.

3. Line 217-219. The authors describe the PCR conditions as "The reactions were performed at 50 °C for 2 min, 95 °C for 2 min, and 94 °C for 45 cycles of 15 s each; 60 °C for 15 s and 72 °C for 20 s, and an additional extension at 72 °C for 5 min. I cannot make sense of this. "94°C for 45 cycles of 15 seconds each"? Why did the authors run the PCR at "94 °C for 45 cycles of 15 seconds each", but not the full program of "denature, anneal and extension"?

4.Figure 2. The figure legend mentions "exposed for 24 hours....exposed for 48 hours....exposed for 72 hours..." (line 246). This description is not clear. It will lead the audience to misunderstand that L. frtunei was continuously exposed in NaDCC for 24, 48, or 72 hours. I think..."24 (48 or 72) hours L. frtunei after conditional exposure" would be better. The same problems occur in other figure legends.

5. Line 266-267. “…the number of cells could not be counted due to the high mortality of animals in this period.” I am curious that the authors showed the SEM results in Figure 2J-L but then mentioned that L. frtunei has a high mortality in this period which made them unable to analyze the number of mucus cells. Here are 2 questions. 1. If they could analyze the gill with an SEM microscope, why did they not have samples to analyze the mucus cell number? 2. The authors mentioned “high mortality”, please show the mortality in the manuscript.

6. Line 317. The sentence described “…after 24 and 48 hours of exposure (Figure 10 B, C, D).” However, the Figure 10D shows the data at 72 hours of exposure. The description does not match the figure citation.

7. Figure 7 and 12. The expression of genes was examined. I am curious about the high standard deviation values. This may be due to the fact that the authors only have a small sample size of each group. I am curious about the significance between sample groups with such high standard deviation values. I would suggest that the authors try to increase the number of samples, which would make these data more robust and convincing.

8. Figure 3, 5A, 9. The font size is too small, which is not easy to read through, please add the solid line to the X and Y axis.

9.The authors have labeled the figure panel in upper case, but the figure legends are in lower case. Please show them consistently. 

Round 2

Reviewer 3 Report

No further comments.